# Thermodynamic and Microstructural Analysis of Lead-Free Machining Aluminium Alloys with Indium and Bismuth Additions

**DOI:** 10.3390/ma16186241

**Published:** 2023-09-16

**Authors:** Simon Rečnik, Maja Vončina, Aleš Nagode, Jožef Medved

**Affiliations:** 1Impol 2000, d. d., Partizanska 38, SI-2310 Slovenska Bistrica, Slovenia; simon.recnik@impol.si; 2Faculty of Natural Sciences and Engineering, University of Ljubljana, Aškerčeva Cesta 12, SI-1000 Ljubljana, Slovenia; maja.voncina@ntf.uni-lj.si (M.V.); ales.nagode@ntf.uni-lj.si (A.N.)

**Keywords:** EN AW 6026, EN AW 1370, indium, thermodynamic calculation, microstructure, low-melting phases

## Abstract

The present study comprises an investigation involving thermodynamic analysis, microstructural characterisation, and a comparative examination of the solidification sequence in two different aluminium alloys: EN AW 6026 and EN AW 1370. These alloys were modified through the addition of pure indium and a master alloy consisting of indium and bismuth. The aim of this experiment was to evaluate the potential suitability of indium, either alone or in combination with bismuth, as a substitute for toxic lead in free-machining aluminium alloys. Thermodynamic analysis was carried out using Thermo-Calc TCAL-6 software, supplemented by differential scanning calorimetry (DSC) experiments. The microstructure of these modified alloys was characterised using SEM–EDS analysis. The results provide valuable insights into the formation of different phases and eutectics within the alloys studied. The results represent an important contribution to the development of innovative, lead-free aluminium alloys suitable for machining processes, especially for use in automatic CNC cutting machines. One of the most important findings of this research is the promising suitability of indium as a viable alternative to lead. This potential stems from indium’s ability to avoid interactions with other alloying elements and its tendency to solidify as homogeneously distributed particles with a low melting point. In contrast, the addition of bismuth does not improve the machinability of magnesium-containing aluminium alloys. This is primarily due to their interaction, which leads to the formation of the Mg_3_Bi_2_ phase, which solidifies as a eutectic with a high melting point. Consequently, the presence of bismuth appears to have a detrimental effect on the machining properties of the alloy when magnesium is present in the composition.

## 1. Introduction

The 6XXX series of aluminium alloys are precipitation-hardening alloys known for their moderate strength, favourable strength-to-density ratio, good ductility, weldability, and corrosion resistance. However, machining these alloys can be challenging due to the tendency to form long, continuous strips or curls that may wrap around the workpiece or cutting tool, causing disruptions in the machining process [1,2,3,4]. To address this issue, the 6XXX series includes a specific group of alloys known as free-cutting or free-machining alloys. These alloys are exceptionally suitable for use with automatic CNC cutting machines [5]. They are characterized by containing a small amount of soft and non-abrasive microstructural constituents that facilitate chip breaking [6]. Lead was traditionally the primary alloying element employed to enhance machinability until international guidelines were implemented both in the European Union and globally, restricting the use of lead to a maximum of 0.4 wt.% due to its toxic nature [7,8]. In recent years, even more stringent regulations have come into effect, compelling manufacturers to develop new free-cutting aluminium alloys without lead content. To replace lead in free-machining aluminium alloys, the substitute element should possess similar properties. These include insolubility in the aluminium matrix and the ability to form phases with low melting points and lower hardness compared to the aluminium matrix [9]. According to these criteria, the alloying elements that can serve as lead replacements are tin, bismuth, cadmium, thallium, antimony, indium, selenium, and mercury. While various elements can enhance machinability, the most commonly used alternatives are tin or bismuth individually or combinations of tin and bismuth [10]. Other elements are typically excluded for either economic reasons (e.g., indium) or due to their toxicity (thallium, antimony, selenium, cadmium, and mercury). Due to the generation of frictional heat during machining, the machined zone can reach temperatures as high as 350 °C or even higher. Consequently, the most crucial attribute of a potential replacement element or a combination thereof is the ability to form low melting point phases or eutectics that liquefy or soften below 350 °C. This not only improves chip breaking but also prevents material buildup on the cutting edge [11,12,13]. Over the last two decades, significant research efforts have been directed towards the development of new lead-free aluminium alloys. Numerous research papers have investigated the impact of tin and bismuth additions on the microstructure and machining properties of 6XXX lead-free alloys. Initially, tin appeared to be a promising substitute for lead in the 6XXX series. Unfortunately, the tendency to hot cracking increases significantly with higher tin content [10,14,15,16,17]. Faltus et al. [18] established that the quantity and type of particles formed through the interaction of bismuth and tin are crucial factors affecting the alloy’s machinability and ductility. With a higher magnesium content and a lower Si/Mg ratio, there is more formation of Mg_2_Sn and Mg_3_Bi_2_ phase particles, which reduces ductility and machinability. Conversely, with a higher Si/Mg ratio and a lower magnesium content, more low melting point eutectics are formed, resulting in improved machinability. In one of our previous studies [19], we also discovered that bismuth and tin interact with magnesium to form Mg_2_Sn, Mg_2_(Si,Sn), and Mg_3_Bi_2_ phases, which decrease the ductility of the EN AW 6060 alloy. Bismuth exhibits a higher affinity to magnesium than to silicon, resulting in the formation of the intermetallic α-Mg_3_Bi_2_ phase during solidification. This phase manifests as angular or hexagonal particles within the α-Al crystal grains or as a binary eutectic (α-Al + α-Mg_3_Bi_2_) with a high melting point [20]. 

Some patented alloys incorporate tin, bismuth, and indium as substitutes for lead in various combinations, with the indium content not exceeding 0.4 wt.% [21,22]. In our study, however, we introduced indium into the alloys EN AW 6026 and EN AW 1370 at concentrations of 0.35 wt.% and 1 wt.%, respectively. The objective of this experiment was to assess the potential viability of indium, either alone or in combination with bismuth, as a substitute for toxic lead in free-machining aluminium alloys. This was carried out to investigate whether a low-melting-point phase formed in the microstructure of these alloys (*T_M_* < 350 °C) could potentially improve machinability. We also performed detailed thermodynamic analyses and microstructural characterizations to clarify the solidification sequence and microstructure evolution. 

## 2. Materials and Methods

The studied free-machining alloys were made from the commercially produced alloy EN AW 6026 (Impol d.d., Slovenska Bistrica, Slovenia, chemical composition in Table 1), 99.995 wt.% pure indium (supplied by Santech Materials, Changsha, China), and 99.99 wt.% pure bismuth (supplied by Wogen, London, UK). For the alloy containing both elements, we first prepared a master alloy consisting of 65 wt.% bismuth and 35 wt.% indium. This composition was selected based on the Bi–In phase diagram, which indicates the stable formation of the BiIn phase at 109.7 °C [23]. For the comparison of the solidification sequence, a commercially available technically pure aluminium alloy EN AW 1370 (Impol d.d., chemical composition in Table 2) was alloyed with the same proportions of indium and the Bi–In master alloy. The sample designations and the respective content of added alloying elements are outlined in Table 3.

When the temperature in the furnace reached 750 °C, indium or the Bi–In master alloy was introduced. Temperature control was achieved using a K-type thermocouple, so that a constant casting temperature of about 750 °C was maintained during all casting operations. Before casting, the melt was manually stirred for about 1 min to prevent the settling of indium or BiIn phases, which possess significantly higher densities than the base alloy. The casting process itself was conducted within a steel mould. The chemical composition of the alloys was determined using an ARL 4460 OES Thermo Scientific spectrometer (Waltham, MA, USA), except for the indium content, which was analysed via X-ray fluorescence spectroscopy (XRF), utilizing the Precious Metals Database. Microstructural analysis was performed using a Thermo Scientific Quattro S field emission scanning electron microscope (FEG SEM) equipped with an EDS SSD Ultim^®^ Max detector from Oxford Instruments. The electron beam was operated at an accelerating voltage of 15 kV and backscattered electrons were used for imaging. Thermodynamic calculations were carried out using Thermo-Calc software from Thermo-Calc in Stockholm, Sweden, while differential scanning calorimetry was performed using the Netzsch STA 449C Jupiter instrument (Mt. Juliet, TN, USA). These methods were utilized to determine phase equilibria and characteristic solidification temperatures.

## 3. Results and Discussion

### 3.1. Chemical Composition of Fabricated Alloys

The chemical composition of all synthesized alloys is listed in Table 4. An indium content of 1.04 wt.% was measured in sample EN AW 1370-In, 0.66 wt.% Bi and 0.37 wt.% In were measured in sample EN AW 1370-BiIn, 1.08 wt.% In was measured in sample EN AW 6026-In, and 0.68 wt.% Bi and 0.32 wt.% In were measured in sample EN AW 6026-BiIn. 

### 3.2. Thermodynamic Calculations

Equilibrium isopleth diagrams and non-equilibrium Scheil–Gulliver solidification models were created using Thermo-Calc software for all synthesised alloys. However, in this article, we present the diagram only for the EN AW 6026-In alloy. These diagrams allow us to identify individual microstructural phases and compare the equilibrium and non-equilibrium states with the actual solidification sequence of the alloy. The Thermo-Calc software calculates these diagrams based on thermodynamic laws, a TCAL-6 database of experimentally determined parameters and the assumption of equilibrium solidification [24]. Table 5 shows the predicted phases for all four alloys. According to all Thermo-Calc calculations, indium is expected not to react with other elements and to be present in its elemental form in the microstructure. A similar prediction is made for bismuth in the alloy EN AW 6026-BiIn. The Thermo-Calc calculations did not predict the formation of the intermetallic phase Mg_3_Bi_2_, which formed experimentally in the alloy EN AW 6026 with bismuth addition and in the alloy EN AW 6060 with tin and bismuth addition [19,20].

Figure 1 and Figure 2b show the isopleth diagram and the non-equilibrium Scheil–Gulliver diagram for the alloy EN AW 6026-In. The solidification of the alloy starts at a liquidus temperature of 645.4 °C with the solidification of primary α-Al crystals. In the initial phase of solidification, in addition to the primary α-Al crystals, a smaller part of the aluminium melt (LIQUID) rich in indium also appears. The solidification sequence continues with the crystallisation of the phases Al_15_Si_2_(Fe,Mn)_4_, Mg_2_Si, β-Si, and Al_13_Cr_4_Si_4_. Due to the altered solubility of the elements in the Al_15_Si_2_(Fe,Mn)_4_ and Mg_2_Si phases, a transformation of the phases occurs at temperatures below 400 °C (Figure 2a), leading to the formation of new phases, such as Al_9_Fe_2_Si_2_ from Al_15_Si_2_(Fe,Mn)_4_ and Al_5_Cu_2_Mg_8_Si_6_ from Mg_2_Si. 

At a temperature of 155.4 °C, indium finally solidifies from the remaining melt in elemental form. According to the equilibrium diagram, the following phases should be present at room temperature: α-Al, Al_13_Cr_4_Si_4_, Al_15_Si_2_(Fe,Mn)_4_, Al_9_Fe_2_Si_2_, β-Si, Mg_2_Si, Al_5_Cu_2_Mg_8_Si_6_, and elemental indium particles. Scheil’s calculation predicts the formation of these phases, with the difference that HCP_A3 is present (Figure 2b). HCP_A3 stands for the interaction between indium and zinc. 

### 3.3. Microstructural Characterization

Based on the mass fractions of the individual elements (SEM–EDS, Figure 3) of constituents in the microstructure of the alloy EN AW 1370-In, the presence of α-Al, Al_13_Fe_4_ phases, and indium in elemental form (Figure 3, Spectrum 2) can be confirmed. The small amount of aluminium detected via EDS analysis in Spectrum 2 could be due to the excitation of the X-rays by the aluminium matrix due to the interaction volume or contamination of the analysed site with aluminium from the matrix during metallographic sample preparation. The presence of elemental indium particles was further validated through a DSC heating analysis, showing an endothermic peak at a temperature of 160.7 °C, which corresponds to the melting point of pure In (Figure 4). Other endothermic peaks appeared at temperatures 641.6 °C, 653.8 °C, and 671.9 °C. The peaks at 641.8 °C and 653.8 °C are attributed to the melting of the eutectics (α-Al + Al_13_Fe_4_ + (Al,In)) and (α-Al + Al_13_Fe_4_), respectively. Small indium particles were detected in some Al_13_Fe_4_ phases (Figure 5). Scheil’s calculation predicted the formation of a metastable phase BCT_A5 resulting from an interaction between Al and In. The endothermic peak at 641.6 °C observed on the DSC heating curve indicates the presence of a ternary eutectic (α-Al + Al_13_Fe_4_ + (Al,In)). The melting point of the α-Al solid solution was determined to be 671.9 °C.

The microstructure of the alloy EN AW 1370-BiIn is shown in Figure 6. Compared to the microstructure of EN AW 1370-In, a higher proportion of spherical eutectics can be observed. From the SEM–EDS analysis, the presence of α-Al, Al_13_Fe_4_, and bright BiIn phases in the microstructure is evident (Figure 7).

The microstructure of the alloy EN AW 1370-BiIn with marked EDS analysis sites is shown in Figure 7 and Figure 8. The results of the EDS analysis confirm the presence of the Al_13_Fe_4_ phase (Figure 7, Spectrum 1) and the BiIn phase (Figure 7, Spectrum 2). The DSC analysis (Figure 9) also detected the endothermic peak at 113.7 °C, which is very close to the BiIn phase melting point (109.7 °C) in the equilibrium Bi–In phase diagram [23]. Figure 8 shows spherical particle consisting of extremely small, bright particles based on Bi, In and Al, combined with Fe and a small amount of Si. According to thermodynamic calculations, the iron phase with Si could be either one of the metastable or stable phases Al_8_Fe_2_Si or Al_8_Fe_2_Si_2_ and the phase based on Bi, Al, and In could be the BCT_A5 phase. The combination of these phases with the α-Al phase could form a ternary eutectic (α-Al + Al_8_Fe_2_Si + (Al, In, Bi)). The DSC curve (Figure 9) shows clear endothermic peaks at temperatures of 646.4 °C, 658.5 °C, and 683.7 °C. These peaks correspond to the melting of the eutectics (α-Al + Al_8_Fe_2_Si + (Al, In, Bi)), (α-Al + Al_13_Fe_4_), and α-Al solid solution.

The alloy EN AW 6026 contains magnesium and silicon as the main alloying elements, together with other trace alloying elements such as Fe, Cu, Mn, Cr, Zn, and Ti. Figure 10 shows the microstructure of the alloy EN AW 6026-In. Based on the mass fractions of each element and the Thermo-Calc calculations, we identified the following microstructural constituents in the microstructure: α-Al, Al_15_Si_2_(FeMn)_3_, Mg_2_Si, Al_5_Cu_2_Mg_8_Si_6_, and In. The EDS analysis confirmed that indium solidified in elemental form (marked as Spectrum 1 and Spectrum 2 in Figure 10). The low enthalpy of indium resulted in the absence of a prominent endothermic peak on the DSC curve (Figure 11). A spherical eutectic was also observed in the microstructure of the alloy, along with smaller In particles (Figure 12). The EDS analysis detected the presence of Al, Mg, Si, and Cu in the constituents of the spherical particle. Scheil’s calculation and isopleth phase diagram for the alloy EN AW 6026-In predicted the solidification of the Al_5_Cu_2_Mg_8_Si_6_ phase. Since we detected all listed elements in the EDS spectrum, we assume that this phase is present in the microstructure. On the DSC heating curve, the first prominent endothermic peak was seen at a temperature of 532.4 °C, indicating the melting of the (α-Al + Al_5_Cu_2_Mg_8_Si_6_) eutectic. Additional endothermic peaks were observed at temperatures of 557.4 °C and 575.6 °C, corresponding to the predicted melting of the (α-Al + Mg_2_Si) and (α-Al + Al_15_Si_2_(FeMn)_4_) eutectics, respectively. The liquidus temperature of the α-Al solid solution was determined to be 646.8 °C.

Figure 13 shows the microstructure of the alloy EN AW 6026-BiIn. Based on EDS analysis and a combination of equilibrium and non-equilibrium solidification diagrams, we identified the following microstructural phases: α-Al, Al_15_Si_2_(FeMn)_4_, Mg_2_Si, Al_5_Cu_2_Mg_8_Si_6_, Mg_3_Bi_2_, and indium particles. Morphologically, the Al_5_Cu_2_Mg_8_Si_6_ phases exhibited similar characteristics to those of the alloy EN AW 6026-In. EDS mapping was carried out to determine the specific distribution of each alloying element, focusing on Mg_3_Bi_2_ and In particles. Indium was found not to react with any of the alloying elements. However, in the case of Mg_3_Bi_2_ phase particles, an overlap between Bi and Mg was observed (Figure 14). Due to the formation of the Mg_3_Bi_2_ phase, the BiIn phase was not detected in the microstructure, unlike the alloy EN AW 1370-In. The DSC heating curve (Figure 15) did not show any endothermic peaks indicating the presence of the BiIn phase’s presence nor were there any prominent endothermic peaks in the temperature range from 150 °C to 170 °C, indicating the melting of indium. Nevertheless, Spectrum 3 in Figure 13 indicated that the identified proportion of indium was 96.28 wt.%, and the mapping analysis demonstrated that In did not interact with other elements, confirming its presence in its elemental form. The first significant endothermic peak (Figure 15) occurred within the temperature range of 536.1 °C to 545.9 °C, corresponding to the melting of the Al_5_Cu_2_Mg_8_Si_6_) and (α-Al + Mg_2_Si) eutectics. The most notable peaks corresponded to the melting of the (α-Al + α-Mg_3_Bi_2_) and (α-Al + Al_15_Si_2_(FeMn)_4_) eutectics at temperatures of 589.9 °C and 655.2 °C, and the α-Al solid solution melted at a temperature of 663.4 °C.

## 4. Conclusions

This article examines the microstructural and thermodynamic characterisation of laboratory-prepared aluminium alloys EN AW 1370 and EN AW 6026 with the addition of pure indium and an In–Bi master alloy, respectively. The aim of the experiments was the formation of a low-melting phase (*T_M_* < 350 °C), which was found in both alloys as indium particles. 

The microstructure of the alloy EN AW 1370-In consisted of α-Al, Al_13_Fe_4_, and pure indium particles, as confirmed via EDS analysis. Differential scanning calorimetry revealed endothermic peaks corresponding to the melting points of pure indium and two eutectics: ternary (α-Al + Al_13_Fe_4_ + (Al,In)) at 641.8 °C and binary (α-Al + Al_13_Fe_4_) at 653.8 °C, while the α-Al solid solution exhibited a melting point of 671.9 °C.The microstructure of the alloy EN AW 1370-BiIn consisted of α-Al, Al_13_Fe_4_, and a BiIn phase. Thermodynamic calculations suggest the possibility of a ternary eutectic (α-Al + Al_8_Fe_2_Si_2_+ (Al, In, Bi)). On the DSC heating curve, the melting of the BiIn phase was found to occur at 113.7 °C, which is very close to the BiIn phase melting point (109.7 °C) in the equilibrium Bi–In phase diagram. The melting points of the eutectics (α-Al + Al_8_Fe_2_Si_2_ + (Al, In, Bi)) and (α-Al + Al_13_Fe_4_) were detected at 646.4 °C and 658.5 °C, respectively, while the α-Al solid solution exhibited a melting point of 683.7 °C.The microstructure of the alloy EN AW 6026-In exhibited a higher degree of complexity and comprising an α-Al matrix, (α-Al + Al_5_Cu_2_Mg_8_Si_6_), (α-Al + Mg_2_Si), and (α-Al + Al_15_Si_2_(FeMn)_4_) binary eutectics as well as particles of elemental indium. Unfortunately, the DSC method could not reliably detect the melting or solidification of indium particles, due to their small number and the low enthalpy of indium. On the DSC heating curve, the initial prominent endothermic peak was seen at 532.4 °C, indicating melting of the (α-Al + Al_5_Cu_2_Mg_8_Si_6_) eutectic. Other endothermic peaks were observed at temperatures of 557.4 °C and 575.6 °C, indicating the melting of the (α-Al + Mg_2_Si) and (α-Al + Al_15_Si_2_(FeMn)_4_) eutectics, while the α-Al solid solution exhibited a melting point at 646.8 °C.The microstructure of the alloy EN AW 6026-BiIn consisted of the α-Al matrix; (α-Al + Al_5_Cu_2_Mg_8_Si_6_), (α-Al + Mg_2_Si), (α-Al + Mg_3_Bi_2_), and (α-Al + Al_15_Si_2_(FeMn)_4_) binary eutectics; and indium in elemental form. EDS mapping demonstrated that indium formed particles in elemental form and did not interact with other elements, while bismuth reacted with magnesium to form the Mg_3_Bi_2_ phase and prevent the formation of the BiIn phase. The DSC heating curve did not exhibit any endothermic peaks indicating the presence of the BiIn phase nor any prominent endothermic peaks corresponding to the melting of the indium particles. Instead, phase changes were observed at temperatures of 536.1 °C, 545.9 °C, 589.9 °C, and 655.2 °C and were attributed to the melting of the following eutectics (α-Al + Al_5_Cu_2_Mg_8_Si_6_), (α-Al + Mg_2_Si), (α-Al + α-Mg_3_Bi_2_), and (α-Al + Al_15_Si_2_(FeMn)_4_), while the α-Al solid solution melted at 663.4 °C.

## Figures and Tables

**Figure 1 materials-16-06241-f001:**
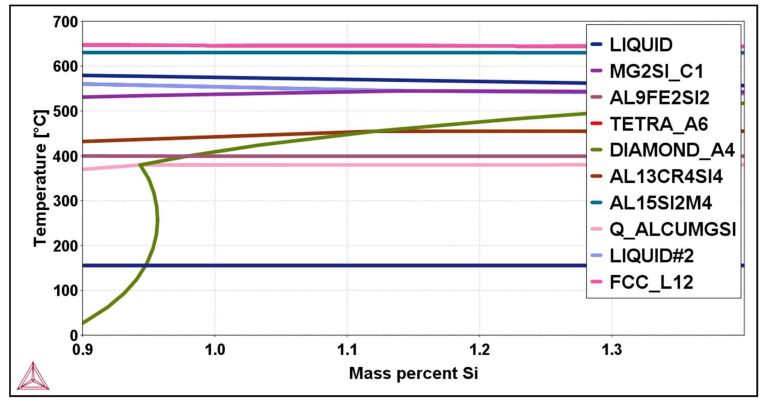
Isopleth phase diagram for EN AW 6026-In.

**Figure 2 materials-16-06241-f002:**
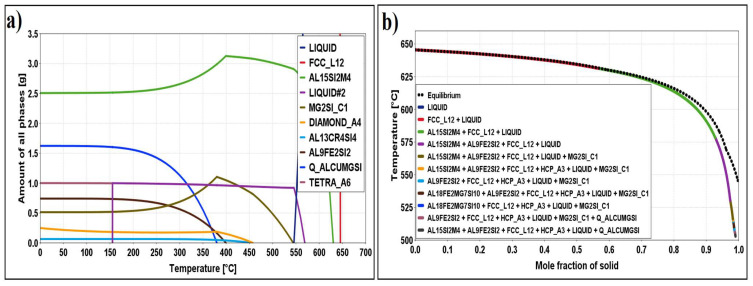
(**a**) Diagram of the proportion of phases as a function of temperature for sample EN AW 6026-In; (**b**) Scheil’s diagram for sample EN AW 6026-In.

**Figure 3 materials-16-06241-f003:**
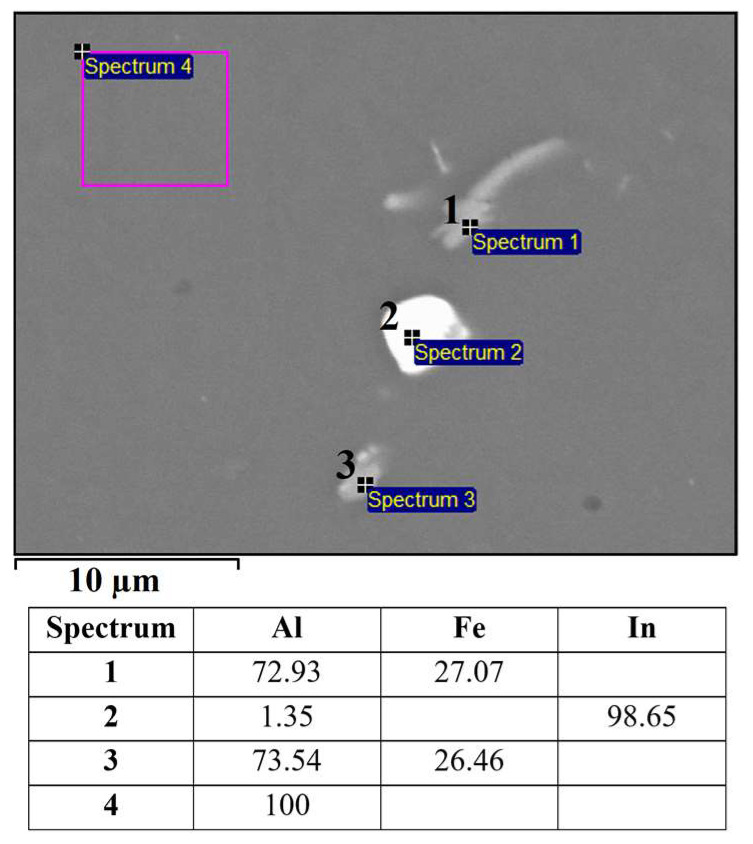
SEM micrograph of EN AW 1370-In with EDS results [wt.%].

**Figure 4 materials-16-06241-f004:**
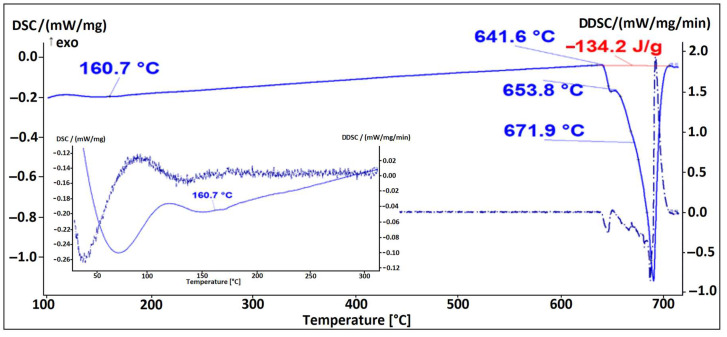
DSC heating curve of the EN AW 1370-In alloy.

**Figure 5 materials-16-06241-f005:**
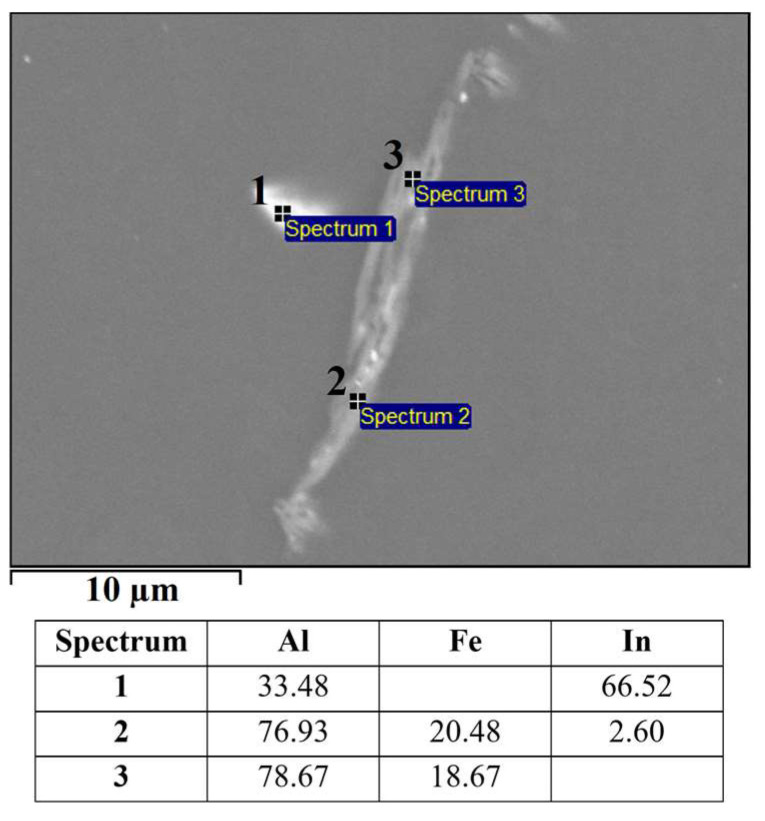
SEM micrograph of ternary eutectic (α-Al + Al_13_Fe_4_ + (Al,In)) with EDS results [wt.%].

**Figure 6 materials-16-06241-f006:**
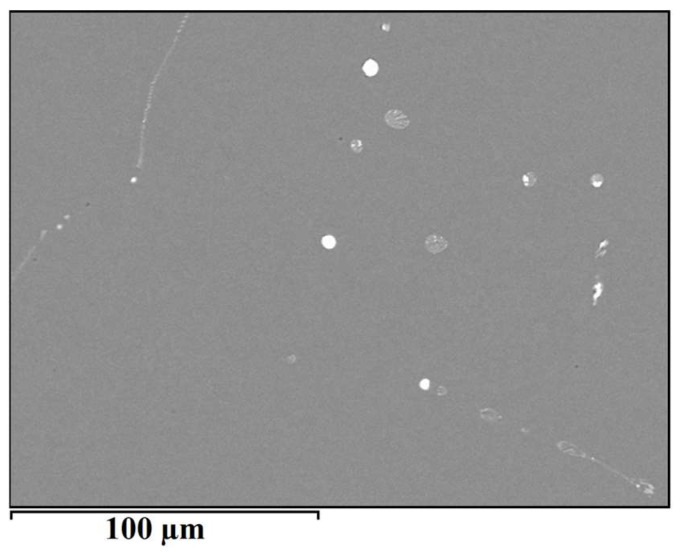
SEM micrograph of EN AW 1370-BiIn alloy.

**Figure 7 materials-16-06241-f007:**
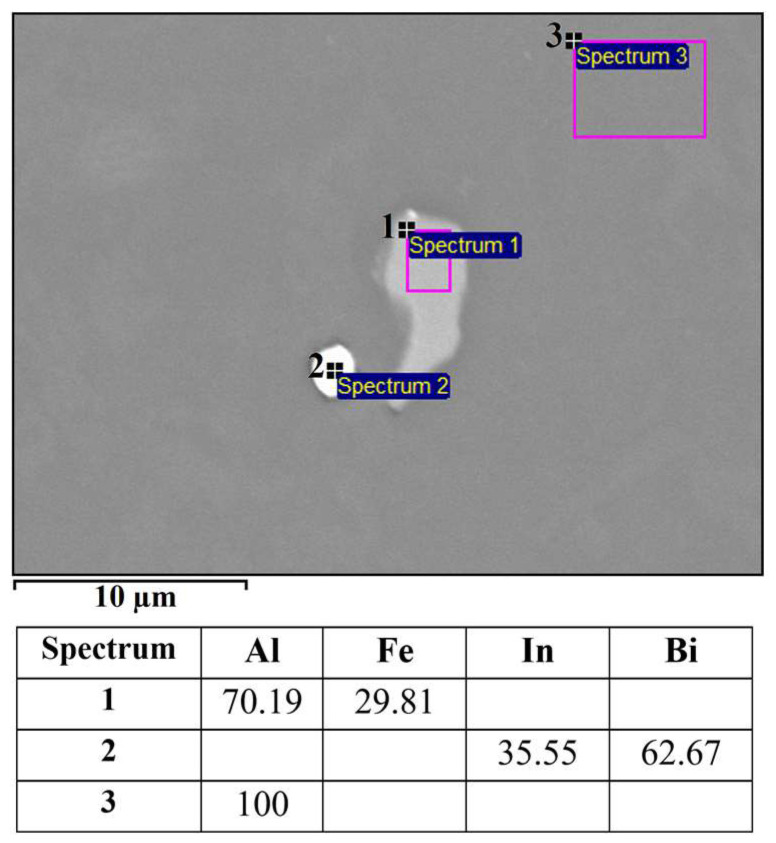
SEM micrograph of EN AW 1370-BiIn with EDS results [wt.%].

**Figure 8 materials-16-06241-f008:**
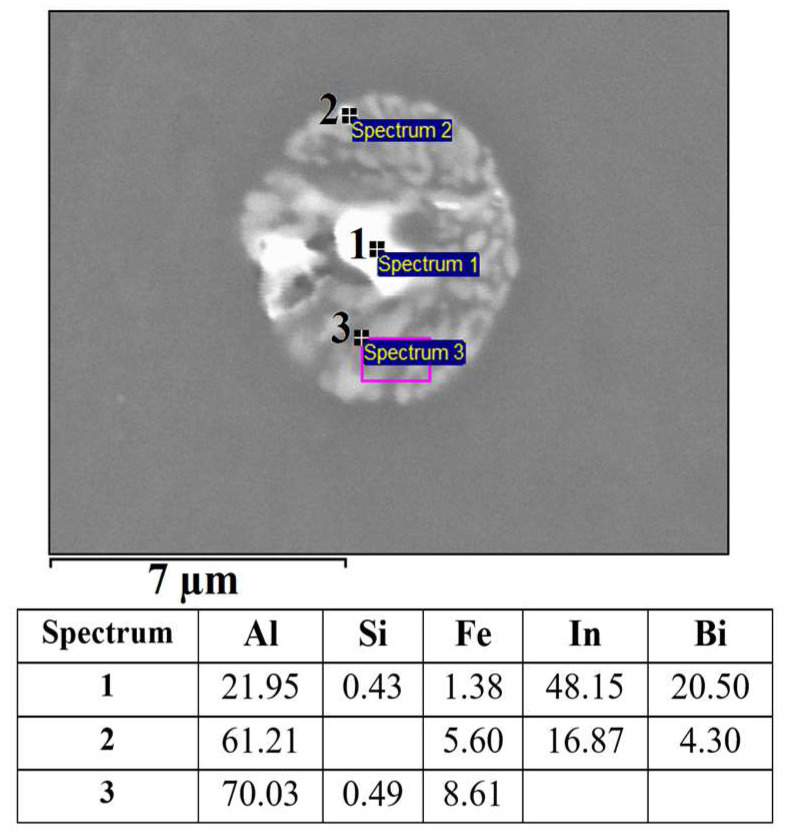
Spherical eutectic particle in the microstructure of the EN AW 1370-BiIn alloy with EDS results [wt.%].

**Figure 9 materials-16-06241-f009:**
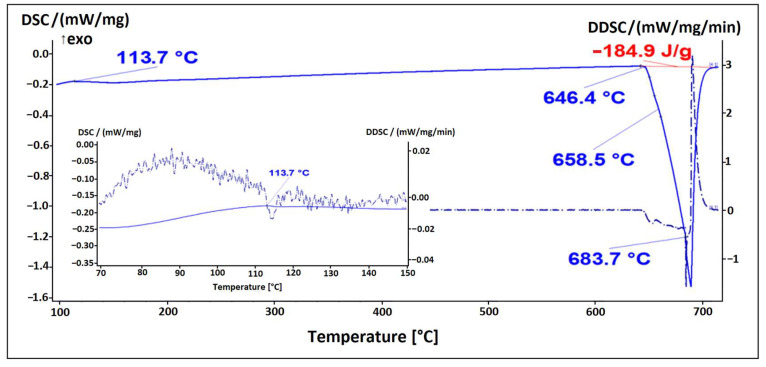
DSC heating curve of sample EN AW 1370-BiIn.

**Figure 10 materials-16-06241-f010:**
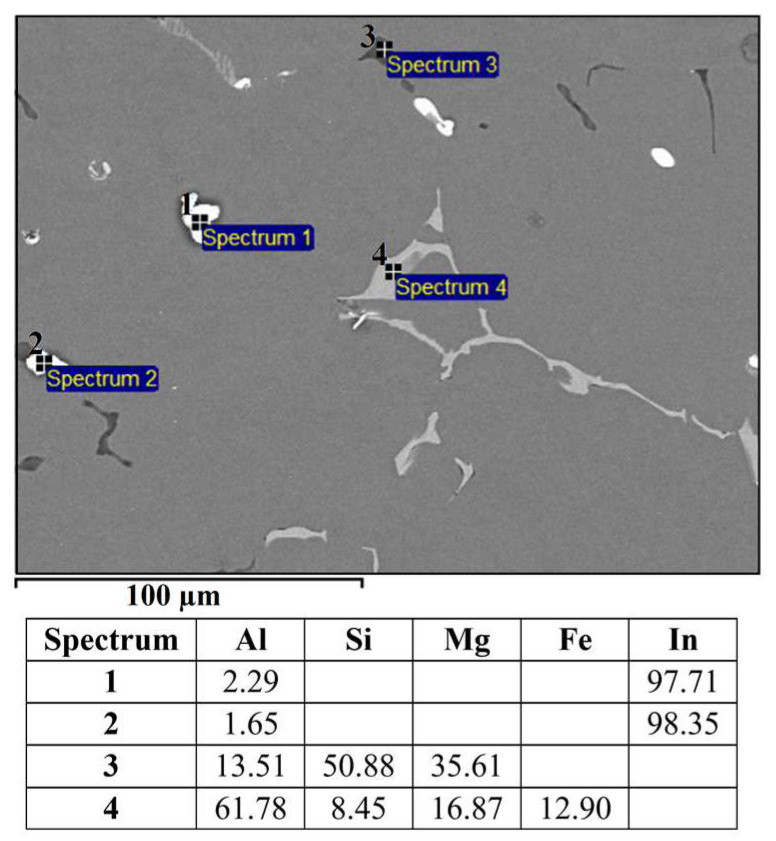
SEM micrograph of the alloy EN AW 6026-In with EDS results [wt.%].

**Figure 11 materials-16-06241-f011:**
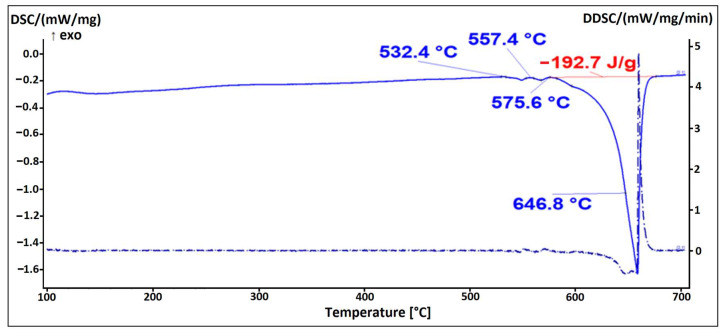
DSC heating curve of the EN AW 6026-In alloy.

**Figure 12 materials-16-06241-f012:**
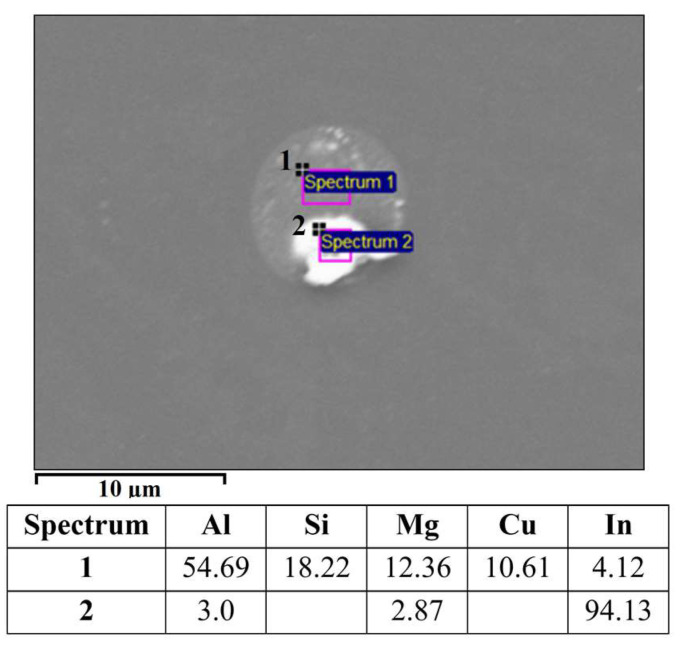
Spherical eutectic in EN AW 6026-In alloy with EDS results [wt.%].

**Figure 13 materials-16-06241-f013:**
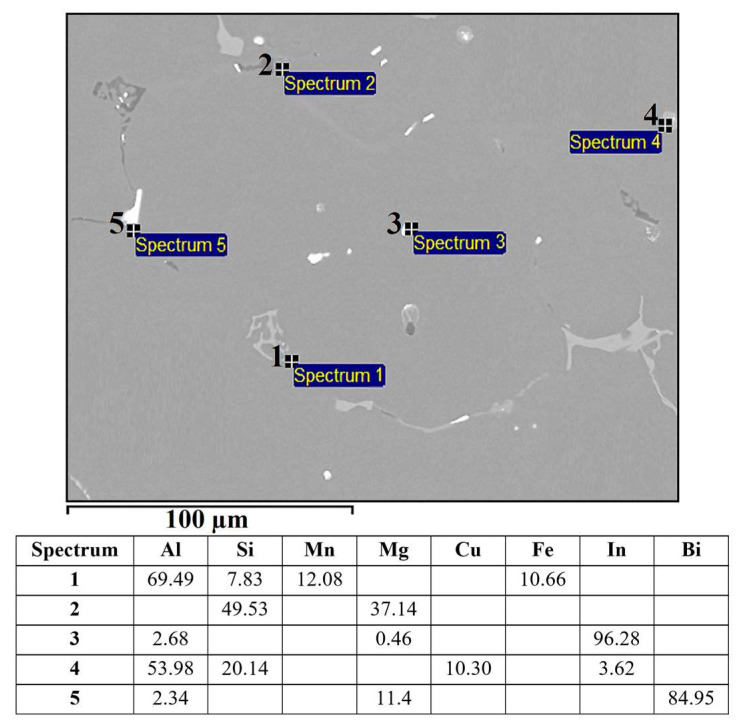
SEM micrograph of the alloy EN AW 6026-BiIn with EDS results [wt.%].

**Figure 14 materials-16-06241-f014:**
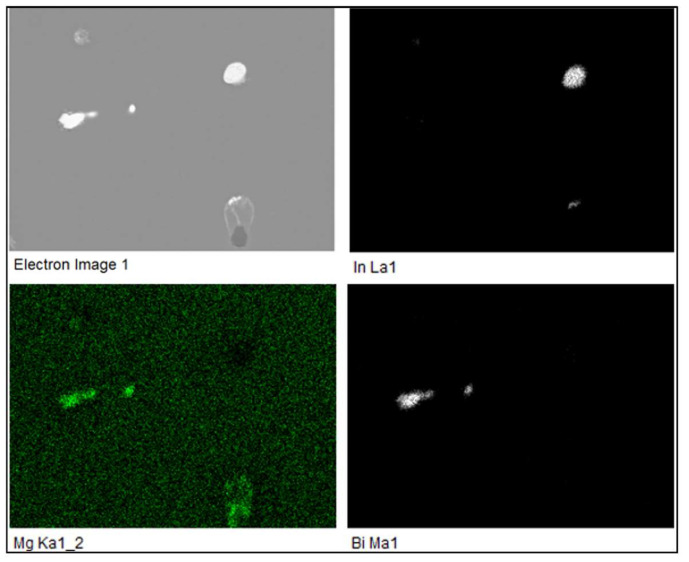
EDS elemental mapping of In, Mg, and Bi in the alloy EN AW 6026-BiIn.

**Figure 15 materials-16-06241-f015:**
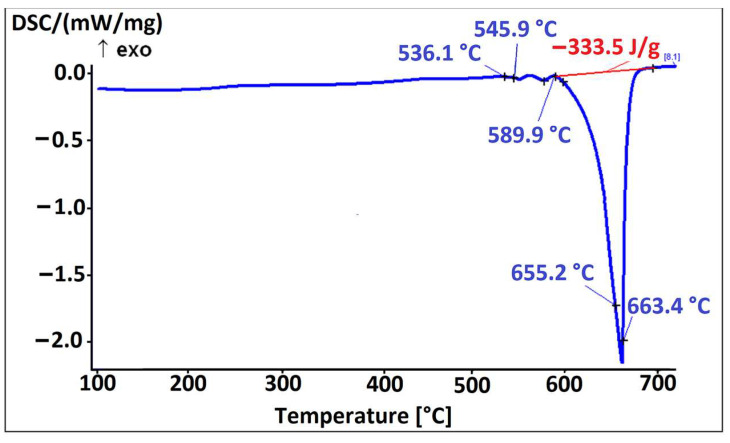
DSC heating curve of the EN AW 6026-BiIn alloy.

**Table 1 materials-16-06241-t001:** Chemical composition of the EN AW 6026 alloy.

Element	Si	Fe	Cu	Mn	Mg	Cr	Zn	Ti	Al
wt.%	1.134	0.201	0.332	0.747	0.825	0.016	0.028	0.017	Bal.

**Table 2 materials-16-06241-t002:** Chemical composition of the EN AW 1370 alloy.

Element	Si	Fe	Cu	Mn	Zn	Ti	Al
wt.%	0.051	0.144	0.047	0.007	0.014	0.006	Bal.

**Table 3 materials-16-06241-t003:** Designation of samples and content of added alloying elements.

Designation	Basic Alloy	Content of Added Element in [wt.%]
EN AW 1370-In	EN AW 1370	1 wt.% In
EN AW 1370-BiIn	EN AW 1370	1 wt.% BiIn
EN AW 6026-In	EN AW 6026	1 wt.% In
EN AW 6026-BiIn	EN AW 6026	1 wt.% BiIn

**Table 4 materials-16-06241-t004:** Chemical composition of synthesized alloys.

Designation	Si	Fe	Cu	Mn	Mg	Cr	Zn	Ti	Bi	In	Al
EN AW 1370-In	0.056	0.167	/	0.007	0.002	0.001	0.009	0.002	/	1.041	Bal.
EN AW 1370-BiIn	0.051	0.142	/	0.006	0.001	0.001	0.007	0.001	0.664	0.373	Bal.
EN AW 6026-In	1.145	0.201	0.319	0.751	0.822	0.015	0.024	0.016	/	1.082	Bal.
EN AW 6026-BiIn	1.113	0.244	0.324	0.774	0.832	0.016	0.021	0.014	0.684	0.321	Bal.

**Table 5 materials-16-06241-t005:** Calculated phases in equilibrium state and Scheil’s calculation with the Thermo-Calc software for all four alloys.

LIQUID = liquid	MG2SI = Mg_2_Si
LIQUID #1 = liquid based on (Al, In, Fe, Si, Mg, ...)	DIAMOND_A4 = β-Si
LIQUID #2 = liquid based on (In, Al, …)	AL13CR4SI4 = Al_13_Cr_4_Si_4_
FCC_L12 = α-Al	AL9FE2SI2 = Al_9_Fe_2_Si_2_
FCC_L12#2 = α-Al (Bi)	Q_AlCUMGSI = Al_5_Cu_2_Mg_8_Si_6_
AL15SI2M4 = Al_15_Si_2_(FeMn)_4_	RHOMBO_A7 = Bi
AL13FE4 = Al_13_Fe_4_	TETRA_A6 = In
AL8FE2SI = Al_8_Fe_2_Si	HCP_A3 = (In,Zn)
AL8FE2SI2 = Al_8_Fe_2_Si_2_	BCT_A5 = (Al,In)

## Data Availability

Not applicable.

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
