# Peer review of "Thermodynamic and Microstructural Analysis of Lead-Free Machining Aluminium Alloys with Indium and Bismuth Additions"

_materials, 2023, doi:10.3390/ma16186241_

Round 1

Reviewer 1 Report

The research paper entitled " Thermodynamic and microstructural analysis of lead-free machining aluminium alloys with indium and bismuth additions (materials-2599550)” was reviewed. After reading the manuscript, I suggest to author to revise wisely for publication in Materials. Please do the revision for this manuscript based on comments below (major revision):

1. The paper contains some grammatical errors and typo-mistakes that should be corrected. The English language should be greatly improved. For better readability of the text, the mathematical/physical symbols should be writing italics, the subscript/superscript notations should be properly rechecked, the equations should be numbered, the units should be rechecked, the illegible/non-understandable axis descriptions should be rechecked, and the abbreviations should be opened when mentioned for the first time, and so on.

2. The abstract should contain useful information such as particle size, charge and effective concentration of microorganisms. Also, it's better to give an introduction sentence in the beginning.

3. Author has to highlight the novelty of Research question very clearly in the introduction which is not mention.

4. The Abstract part should be revised. It should clearly summarize the problem, state the concept and the method, and inform the important results and conclusions in the present study. As well, it should contain some qualitative and quantitative results.

5. The introduction part should be further improved. These materials are very promising materials for several practical applications, which can be highlighted in the Introduction part. So, some recent references should be inserted and discussed which will be very helpful for the researchers/readers. The authors should look at the other studies and discuss them in the manuscript.

6. Fig. 2 and 4 not clear

7. The conclusion part should be more concise. It should summarize the important observations, major findings (in qualitative and quantitative forms), and future perspectives of the present work.

8. The author needs to discuss the obtained result with the previously published data. This manuscript lacks discussion.

Extensive editing of English language required

Author Response

We would like to thank you for your comments and suggestions. We are glad that we were able to improve the manuscript.

We have made corrections in response to your comments:

Response 1: Following your recommendation to fix grammatical errors, we have revised the article and arranged for a review using the Insta-Text programme.

Responses 2 and 4: The abstract of the article has been revised in accordance with your suggestions in points 2 and 4. The updated abstract, now thoroughly communicates the basic concept of the experiment and the objectives of the article and provides extended insights into the main findings.

Response 3 and 5: In response to the reviewer's suggestion, we have made changes to the introduction section of the article. We have also tried to improve the explanation of the research question in this section.

We have also tried to add some more references to the introduction. However, it should be noted that there are only a limited number of articles on this specific research topic.

Response 6: We have undertaken a redesign of Figures 1, 2, 3 and 4. As a result, the images are now clearer and we have also adjusted the font size.

Response 7: In response to the suggestions submitted, we have restructured the conclusions. This revision includes a more comprehensive description of the microstructural characteristics of the alloys and detailed explanations of the phase changes observed in the DSC curves.

Response 8: This article addresses the discussion in the Results and Discussion section. The guidelines for this journal generally allow these two paragraphs to be combined. In our article we have focused specifically on the microstructural characteristics of alloys. We have not found similar previously published data on the addition of indium to aluminium alloys where microstructural characterisation has been performed. Therefore, we have not made comparisons with other work as part of the discussion.

Hoping that the above mentioned changes in the manuscript and our answers satisfy all reviewer comments,

Kind regards.

Reviewer 2 Report

1. In the abstract section, there is a lack of quantitative description.

2. in lines 64-71, the innovation in the introduction section is not clear enough, please describe the innovation better.

3. the text in Figure 1 should be adjusted properly to make the picture look clearer.

4. The third and fourth articles in the conclusion should be streamlined to make it more simple and clear to the reader.

Author Response

We would like to thank you for your comments and suggestions. We are glad that we were able to improve the manuscript.

We have made corrections in response to your comments:

Response 1: The abstract of the article has been revised according to your suggestions. The updated abstract now thoroughly conveys the basic concept of the experiment and the aims of the article, and provides an extended insight into the main results.

Response 2: In response to the reviewer's suggestion, we re-formulate the sentences in lines 64-71.

Response 3: We have undertaken a redesign of Figures 1, 2, 3 and 4. As a result, the images are now clearer and we have also adjusted the font size

Response 4: In response to the proposals submitted, we have re-structured the conclusions. This revision includes a more comprehensive description of the microstructural characteristics of the alloys and detailed explanations of the phase changes observed in the DSC curves.

Response 7: We have removed the term 'spherical eutectic' from the sentence.

Hoping that the above mentioned changes in the manuscript and our answers satisfy all reviewer comments,

Kind regards.

Reviewer 3 Report

The manuscript "Thermodynamic and microstructural analysis of lead-free machining aluminium alloys with indium and bismuth additions" describes obtaining four new lead-free alloys based on aluminium, which are doped by indium and bismuth, or both metals. The alloys meld were studied using scanning electron microscopy, differential scanning calorimetry. EDS and XRF analysis was employed to study the chemical composition of separate grains in alloys. In addition, theoretical calculations of different phases formed in alloys were performed.

Microstructure of alloys were determined, as a result. Melting points related to the different phases were experimentally identified. The results of the paper will come handy for those specialists in material science who intend to replace lead with non-toxic metals.

The only question remains unclear to me after reading this manuscript: how the chemical resilence of the alloys studied have changed after replacing lead by Bi and In? Was it improved or otherwise? The study of the changes in mechanical properties of the Pb-free alloys (hardness, ductility) would also be interesting. Perhaps, Auuthors would pay attention to these questions in their future endeavors.

Author Response

We would like to thank you for your comments and suggestions. We are glad that we were able to improve the manuscript.

We have made corrections in response to your comments:

Response 1: In this article, we have mainly focused on the microstructural characterisation of aluminium alloys. However, we plan to conduct further analyses in future research, where we will investigate how the addition of indium affects the mechanical properties of the alloy itself.

Hoping that the above mentioned changes in the manuscript and our answers satisfy all reviewer comments,

Kind regards.

Reviewer 4 Report

The submitted paper is a well-worked, well-knit and well-presented article. Congratulations to the authors for the achievement. However, it has some questionable areas that need to be corrected::

1. Shaking before pouring (lines 93 and 94) I consider to be the weak link of this article. I personally made a straight Al-Zn alloy with another purpose: to deposit it on the surface of a steel wire in order to protect it against corrosion. The Al-Zn alloy produced depended extremely much on the degree and method of stirring before its casting. In practice, I obtained extremely different materials in this way. Once again, it was Al-Zn alloy with the same chemical composition, but differing only in the degree and method of stirring before casting. It could therefore be that the degree and duration of agitation have a huge impact on the final characteristics.

2. On line 118, instead of the words "...is shown in article..." the grammatically correct form "...are shown in article..." will appear because on line 116 you use the plural: "Equilibrum isopleth diagrams ....”

3. At line 131 of table 5, change the last cell: "HCP_A3=(In,Zn)" as correctly specified in line 146,

4. Replace all figure legends (from fig. 1 to fig. 15) according to the requested template, e.g. instead of:" Figure 1: Isopleth phase diagram for EN AW 6026-In" will appear:" Figure 1: Isopleth phase diagram for EN AW 6026-In"

5. In my opinion, the presentation of figure 6 is pointless, because it observes something without proving (possibly through EDS mapping) that something.

6. At line 204, instead of "...Figures 8 and 9." will be replaced by "... Figures 7 and 8."

7. At line 321 I would delete: "Spherical eutectic particles were also found." because it is an unproven conclusion.

Overall, after the correction, I honestly believe that the article is valuable and will be consulted by a large number of researchers in the field.

Minor editing of English language required

Author Response

We would like to thank you for your comments and suggestions. We are glad that we were able to improve the manuscript.

We have made corrections in response to your comments:

Response 1: We acknowledge your concerns about shaking before casting. Nevertheless, it is common practise to mix the melt before the actual melting process, usually in conjunction with argon bubbling. This procedure is used under industrial conditions and prevents denser particles from settling in the melt.

Response 2: The sentence is now formulated differently and is grammatically correct.

Response 3: The correction in Table 5 has been implemented.

Response 4: The figure captions have been revised according to the template provided.

Response 5: With your permission, we would like to leave Figure 6 in the article. The main purpose of this figure is to illustrate the different microstructural properties of the alloys AW EN 1370-In and AW EN 1370-BiIn. In particular, it visually distinguishes different microstructural components based on phase morphology.

Response 6: We have corrected the numbering of the images.

Response 7: We have removed the term 'spherical eutectic' from the sentence.

Hoping that the above mentioned changes in the manuscript and our answers satisfy all reviewer comments,

Kind regards.

Round 2

Reviewer 1 Report

Accept

Minor editing of English language required